# Parental Perceptions and Exposure to Advertising of Toddler Milk: A Pilot Study with Latino Parents

**DOI:** 10.3390/ijerph18020528

**Published:** 2021-01-10

**Authors:** Emily W. Duffy, Lindsey S. Taillie, Ana Paula C. Richter, Isabella C. A. Higgins, Jennifer L. Harris, Marissa G. Hall

**Affiliations:** 1Department of Nutrition, Gillings School of Global Public Health, University of North Carolina at Chapel Hill, Chapel Hill, NC 27599, USA; ebwelker@email.unc.edu (E.W.D.); taillie@unc.edu (L.S.T.); 2Carolina Population Center, University of North Carolina at Chapel Hill, Chapel Hill, NC 27599, USA; apcr@live.unc.edu (A.P.C.R.); ihiggins@email.unc.edu (I.C.A.H.); 3Department of Health Behavior, University of North Carolina at Chapel Hill Gillings School of Global Public Health, Chapel Hill, NC 27599, USA; 4Rudd Center for Food Policy & Obesity, University of Connecticut, Hartford, CT 06103, USA; jennifer.harris@uconn.edu; 5School of Medicine, UNC Lineberger Comprehensive Cancer Center, University of North Carolina at Chapel Hill, Chapel Hill, NC 27514, USA

**Keywords:** Hispanic Americans, food labeling, child nutrition sciences, pediatric obesity, sugary-sweetened beverage

## Abstract

Marketing of toddler milk (i.e., typically sugar-sweetened nutrient-fortified milk-based drinks marketed for children 12–36 months) is an emerging public health problem in the US. The American Academy of Pediatrics recommends against the consumption of toddler milk because it often contains added sugar and can displace nutrient-dense foods. Studies have not examined toddler milk perceptions among Latinos, an important gap given Latino children in the US are at high risk of having poor diet quality, and toddler milk is extensively advertised on Spanish-language TV. This study used an online survey of a convenience sample of 58 Latino parents to examine parents’ experiences with toddler milk, understand their perceptions of the healthfulness and the nutrition-related claims on toddler milk, and describe their exposure to toddler milk advertising. Nearly half (44%) of parents in the sample reported purchasing toddler milk. When asked to provide open-ended interpretations of claims on toddler milk, almost all parents gave positive answers, suggesting potential “health halo” effects of the claims. More than half (56%) of parents reported seeing toddler milk advertisements, most commonly on Spanish-language TV. The misperceptions about toddler milk identified should be explored in further research using larger, more representative samples.

## 1. Introduction

Diet quality in early childhood is a key determinant of longer-term risk of diet-related chronic diseases [1,2,3]. A key component of diet quality in early childhood is beverage consumption, as beverages contribute a significant proportion of daily energy and key nutrients such as calcium and vitamin D [4]. The only beverages recommended for consumption in early childhood are water, milk, and a limited amount of 100% juice if fruit recommendations cannot be met with whole fruit [5]. As a result of a variety of socioecological determinants such as targeted marketing and acculturative stress [6,7], Latino children often have worse diet quality, including higher consumption of sugary drinks and 100% juice, than non-Latino white children [6,7,8,9,10]. Poor diet quality in early childhood is associated with obesity risk later in childhood, and Latino children in the US are at disproportionate risk of having obesity [11,12]. Public health efforts are urgently needed to improve diet quality and prevent obesity among Latino children in the US.

One troubling trend in early childhood diet quality and beverage consumption in the US is the promotion of toddler milks [13]. Toddler milks are nutrient-fortified milk-based drinks that are typically sugar-sweetened and marketed for children 12–36 months [5,14]. The American Academy of Pediatrics and other major nutrition and health organizations recommend against consumption of toddler milk because it can interfere with sustained breastfeeding, often contains added sugar, offers no unique nutritional value beyond what an adequate diet can provide [5]. Recommendations state that children younger than two years should not consume added sugars because excessive added sugar intake is associated with diet-related diseases (e.g., obesity, type two diabetes, dental caries) and because added sugar intake in early childhood can contribute to sweet preference development [5,15]. Itis also recommended that young children consume a variety of nutrient-dense foods, and toddler milk has the potential to displace these foods. Additionally, the World Health Organization includes toddler milk in its International Code of Marketing of Breastmilk Substitutes (The Code) [16]. The Code, which has not been adopted in the US, calls for the prohibition of marketing these products to the general public, among other provisions [17].

Despite these recommendations, formula companies in the US are focusing on advertising expenditures on toddler milks. Between 2011 and 2015, advertising expenditures on toddler milk in the US increased by 78%, surpassing spending on infant formula advertising expenditures by $7 million in 2015 [18]. Analyses of sales data suggest these increases in advertising expenditures are translating into increases in toddler milk sales in the US. Between 2006 and 2015, volume sales of toddler milks in the US increased from 1 million kg to 3 million kg [13]. There is also evidence of targeted toddler milk marketing to Latino communities. Toddler milk brands are extensively advertised directly to Latino parents on Spanish-language TV [18]. For example, in 2015, Nido (one popular toddler milk brand) spent all of their TV marketing budget ($4 million) on Spanish-language TV, and Enfagrow increased their expenditures on Spanish-language TV from $0 in 2012 to $5 million in 2015 [18]. Little research exists on toddler milk consumption in the US, but one study found that Latino parents were more likely than non-Latino White parents to report serving toddler milk to their children [19].

Toddler milks also carry many nutrition or health-related claims on the front of the package [14]. These claims are often what the US Food and Drug Administration (FDA) considers structure/function claims [14]. This category of claims does not require preapproval by the FDA and does not need to be substantiated by scientific evidence [20,21]. Nutrition and health claims increase parents’ perceived healthfulness and purchase intentions regardless of a food or beverage’s actual nutritional quality [22,23,24,25]. Claims often cause what is called a “health halo”, where shoppers misinterpret a claim about one product attribute to mean the product is generally healthy [26]. Studies have found that parents generally perceive toddler milks to be healthy despite not being recommended [19,27]. One study examined parents’ perceptions of toddler milk claims and found parents believed the claims meant toddler milk provided nutrients other food sources could not provide or that toddler milk was a necessary component of a child’s diet [27]. Another study found that parents’ agreement with claims on toddler milk packaging was associated with increases in the probability of providing toddler milk to their children [19]. However, no studies have explored perceptions of toddler milk claims among Latino populations.

The objectives of this study were to examine Latino parents’ experiences with toddler milk, understand their perceptions of the overall healthfulness of toddler milk and the nutrition-related claims on toddler milk, and describe their exposure to toddler milk advertising using a convenience sample.

## 2. Materials and Methods

### 2.1. Participants

From August to October 2019, we recruited a convenience sample of 61 Latino parents living in North Carolina as part of a pilot study evaluating the impact of sugary drink warnings and taxes on purchases in a naturalistic convenience store laboratory (i.e., the “store pilot study.”). This sample answered questions about toddler milks as part of their participation in the store pilot study, as described below. The results of the store pilot study will be reported in a separate publication.

We recruited participants for the store pilot study in-person (e.g., at bus stops, laundromats, neighborhoods, local nonprofits) via flyers and by word-of-mouth (e.g., hearing about the study from a friend or family member). To be eligible, participants had to be at least 18 years old, identify as Latino or Hispanic, have at least one child (ages 2–18 years), read and speak English or Spanish, do at least half of the grocery shopping for their household, consume sugary drinks at least once in the last month, purchase at least one non-alcoholic beverage in the previous week, and be able to use a computer or tablet to take surveys. Only one person per household could participate. These eligibility criteria were designed for the purpose of the store pilot study.

### 2.2. Procedures

All study participants provided written informed consent. As part of the main pilot study, participants attended five weekly study visits in a naturalistic convenience store laboratory, where they completed a shopping task (data to be reported separately) and a self-administered online survey, which was programmed using Qualtrics survey software. The toddler milk survey items summarized in this manuscript were included in the self-administered online survey from visit three of the store pilot study. At each visit, participants received an incentive totaling $45 ($70 at the fifth visit) in the form of a Visa gift card and grocery items selected during the shopping task. The University of North Carolina Institutional Review Board approved this study. Prior to data collection, this study was pre-registered on AsPredicted.org: http://aspredicted.org/blind.php?x=a92mw3.

### 2.3. Measures

Participants chose to take the survey in English or Spanish. A professional translation company translated survey items from English to Spanish. The visit one survey assessed standard demographic measures. This paper reports the results of the visit three survey items regarding toddler milk. Participants viewed images of two toddler milks that are advertised on Spanish-language TV [18] (Nido Kinder 1+ and Enfagrow Toddler Next Step) and a definition of toddler milk that stated these products are different from infant formulas like Enfamil and Similac (for infants younger than 12 months old) to minimize confusion with infant formula.

The survey assessed familiarity with (i.e., ever seeing toddler milk in a store) and reported purchases of toddler milk (never, 1–2 times, 3–4 times, 5–9 times, 10 or more times). Participants were asked why other parents would want their children to drink toddler milk as we were unsure if a substantial proportion of parents would have purchased toddler milk themselves. The survey also assessed exposure to toddler milk advertising and conversations about toddler milk [28].

Participants then viewed an image of Nido Kinder 1+ and responded to questions about familiarity, past purchases, perceived healthfulness (5-point Likert scale from 1 unhealthy to 5 healthy) [29], and perceptions of added sugar content (Nido Kinder 1+ contains added sugar). Two open-ended items assessed interpretations of structure/function claims (“Helps support healthy growth” and “immunity”) on the Nido Kinder 1+ package. Exact item wording for all measures available in Appendix A. Nido Kinder 1+ was selected as the brand for the image and items because of Nido’s extensive marketing to Latino communities and because Nido Kinder 1+ contains added sugar. These items were developed by the study team, which includes individuals with policy and legal expertise related to toddler milks, using some items that were modified from prior studies (i.e., items related to conversations and perceived healthfulness) and some new items designed to address gaps in the existing literature.

### 2.4. Analysis

Our analytic sample included 58 parents; two parents withdrew from the study before completing the visit three surveys, and one parent did not attend visit three. Descriptive statistics assessed toddler milk familiarity, purchases, other parents’ reasons for provision, perceived healthfulness, advertising exposure, and conversations. For the interpretations of claims, we used an inductive coding approach to develop a set of themes after reviewing participants’ responses. We then used these themes to develop our codebook that contained the name of each theme, a description of the theme, and examples of quotes that would and would not be coded under each theme. We then coded participants’ open-ended responses into themes. Prior to coding, a fluent Spanish speaker (IH) translated all Spanish responses to English. Two coders (ED and AR) double coded all responses, with discrepancies resolved by coder consensus. Due to the small sample size of this pilot study, all data presented are descriptive, and no statistical tests were conducted. Analyses used Stata version 16.1.

## 3. Results

### 3.1. Demographic Characteristics

Participants had a mean age of 35.8 years, and 98% were female (Table 1). About one-quarter (24%) had a child younger than three years in their household. About one-third of parents (39%) had less than a high school degree, and 52% of parents had a high school degree. Most parents (82%) had an annual household income of less than $25,000. Most participants (77%) were overweight or obese. Most participants completed the surveys in Spanish (83%).

### 3.2. Familiarity and Purchase Behaviors

Almost all parents (93%) were familiar with toddler milk. About half of parents (56%) reported never purchasing any toddler milk brand, while 23% had purchased it one to nine times, and 21% had purchased toddler milk 10 or more times (Table 1). When asked about Nido specifically, 98% (57) of parents were familiar with Nido, and 51% (29) reported previously purchasing it. When asked why they thought parents would want their children to drink toddler milk, common reasons were to provide nutrients (72%), to support growth (52%), to help with brain development (41%), because they grew up drinking toddler milk (41%), and because the child likes the taste (28%) (Figure 1).

### 3.3. Perceptions of Healthfulness and Product Claims

About one in four parents (28%, 16) incorrectly stated Nido Kinder 1+ did not contain added sugars. Thirty-nine percent (21) believed it would be healthy for a child to drink toddler milk every day (15% answering 4 and 24% answering5 on a 5-point scale), and only 22% (12) of parents said it would be unhealthy (2% answering 1 and 20% answering 2 on a 5-point scale). When asked to provide an open-ended explanation of their healthfulness rating, common responses for healthy ratings included that toddler milk contained vitamins or other beneficial ingredients, it was a healthy product, and that the participant had consumed toddler milk as a child.

When shown an image of Nido Kinder 1+ with the claim “Helps support healthy growth” and asked, “What does the phrase “Helps support healthy growth” tell you about the product?”, 40% of parents mentioned something directly related to growth, such as “It helps the growth of the child” (Table 2). However, many parents interpreted the claim more broadly: 35% indicated it meant that Nido contained vitamins, minerals, or other nutrients, and 13% thought it meant Nido supported children’s development. When shown an image of Nido Kinder 1+ with the claim “immunity” and asked, “What does the word “immunity” tell you about the product?” most parents (67%) thought it meant the product prevented illness or boosted the immune system (Table 3). Seven percent of parents stated the claim meant the product contained vitamins, minerals, or other nutrients (“That their immune systems are reinforced by the vitamins and minerals”). Few (4%) parents expressed skepticism about either claim (Table 2 and Table 3).

The prevalence of the themes: bone or muscle, meal or milk substitute, other ingredients, and brain or cognition in parents’ interpretations of the “immunity” claim was 0%, so they are not presented in this table.

### 3.4. Advertising Exposure and Conversations

About half of parents (53%, 31) reported having seen toddler milk advertisements, including on Spanish-language TV (68%, 21), supermarkets (55%, 17), social media (35%, 11), coupons (26%, 8), and retailer websites such as Amazon or Wal-Mart (26%, 8). Less common advertising outlets included magazines (19%, 6), English-language TV (16%, 5), convenience stores or gas stations (16%, 5), and parenting websites (13%, 4). Parents reported seeing advertisements for Nestle/Nido most often.

Among parents who had purchased toddler milk (44%, 25), most (88%, 22) reported having at least one conversation related to toddler milk, including with a family member other than their spouse (55% of those who had a conversation,12), health care providers (50%, 11), spouses (32%, 7), friends (27%, 6), and children (23%, 5).

## 4. Discussion

In our study, nearly all Latino parents were familiar with toddler milk and few perceived it to be an unhealthy product, consistent with prior studies [19,27]. Furthermore, more than one-quarter incorrectly believed toddler milk does not contain added sugars. A common reason mentioned for why parents believed toddler milk was healthy was because they grew up drinking it, suggesting that cultural norms may play a role in the provision of toddler milk in Latino communities. Future studies could examine this finding further using qualitative methods and examine differences by acculturation status and Hispanic country of origin as The Code is only enforced in five Latin American countries [30]. Additionally, many parents (44%) in our study reported purchasing toddler milk at least once in the past, but 56% reported they had never purchased toddler milk. These findings should be explored further using larger samples, and food purchasing data and parent demographic characteristics associated with regular toddler milk purchases should be examined.

Few parents expressed skepticism about toddler milk packaging claims, also consistent with studies finding that parents generally agree with toddler milk marketing claims [19,27]. Parents had broad interpretations of a claim about healthy growth, stating that they thought the claim meant that toddler milk contained vitamins and helped with brain development. This finding suggests this claim may create a “health halo” effect (i.e., in which consumers interpret a claim about one product attribute to mean the product is generally healthy) as these attributes were not explicitly mentioned in the claim [26]. Parents also generally had favorable interpretations of a claim about immunity, with many stating the claim meant toddler milk would prevent illness or that it contained vitamins to prevent illness. These responses are in line with evidence that immunity claims mislead consumers, which has led to action by the Federal Trade Commission (FTC) against food companies for using claims about immunity [31]. In addition to further FTC action to prevent misleading marketing, experts have called for FDA to issue industry guidance to limit the use of structure/function claims on toddler milk, similar to what they have proposed for claims on infant formula. Additionally, Congressional action could require the FDA to create a new regulatory framework specifically for claims on products marketed to children 36 months and younger [32]. This framework could require that foods and beverages meet specific nutrition standards to carry claims [32].

About half of the parents in our study reported exposure to toddler milk advertisements, most commonly on Spanish-language TV, and the most common advertised brand was Nestle/Nido. Aggressive toddler milk advertising in the US has contributed to sales increases nationwide [13], and in 2015 Nido spent $4 million on TV advertising, exclusively on Spanish-language TV [18]. Additionally, Enfagrow did not advertise toddler milk on Spanish-language TV prior to 2012 but increased their advertising expenditures to more than $5 million on Spanish-language TV by 2015 [18]. In addition to TV advertising, many parents reported seeing advertisements for toddler milk in supermarkets and on social media. Formula companies purchase consumer data such as whether or not mothers have created a baby registry, posted on Facebook about a pregnancy, or purchased baby items to target parents with advertisements on social media platforms [33], so it is plausible these tactics are also employed for toddler milk advertising. Future research could explore the extent to which Latino parents may be targeted by toddler milk advertisements in other avenues such as in-store marketing (e.g., end cap displays) in Latino grocery stores or on social media platforms.

A strength of this study is its primarily Spanish-speaking Latino sample. Although this was a small convenience sample of parents that are not representative of the larger US Latino population, and some of whom did not currently have a child less than three in the household, both of which limit the external validity of this study. Additionally, some parents viewed a sugary drink warning or tax in the pilot study, which could have reduced their toddler milk healthfulness perceptions. The use of some survey items that have not been validated or tested for reliability is also a limitation. We asked parents why they believed other parents would want their children to drink toddler milk, not knowing how common toddler milk provision would be among our sample. However, future studies should explore parents’ own reasons for providing toddler milk. In addition, participants needed to be comfortable using a computer to participate in the study, potentially excluding some parents with limited computer literacy. Some parents may have confused toddler milk with infant formula, potentially leading to measurement error. However, multiple steps were taken to reduce confusion with infant formula, such as providing a clear definition of toddler milk at the start of the survey as well as the use of images of Nido, a brand that does not produce infant formula, as the example of toddler milk throughout the survey. Finally, the results presented on parents’ interpretations of claims cannot be generalized to all brands of toddler milks.

## 5. Conclusions

The misperceptions about toddler milk identified in this study should be addressed through future research that may inform policy changes such as action by Congress, the FTC, or FDA to prevent misleading labeling of toddler milk and targeted marketing of toddler milk to Latino communities [14,32]. Additionally, some parents reported discussing toddler milk with health care providers. This finding should be explored further, as other studies have documented formula companies’ use of health workers and healthcare settings to promote their products [17]. Finally, given the high percentage of our sample that reported previously purchasing toddler milk, it will be important to use nationally representative studies such as the National Health and Nutrition Examination Survey to understand if many Latino children are consuming toddler milk regularly and if there are disparities in toddler milk consumption in the US.

## Figures and Tables

**Figure 1 ijerph-18-00528-f001:**
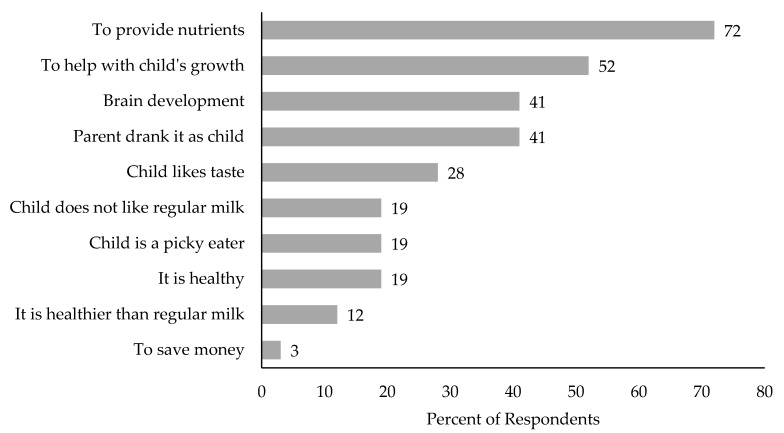
Reasons reported why other parents would want their children to drink toddler milk (*n* = 58).

**Table 1 ijerph-18-00528-t001:** Participant demographic characteristics (*n* = 58).

		Overall	
		N	%
**Age**			
	18–29 years	11	19
	30–39 years	30	52
	40–49 years	16	28
	50+ years	1	2
	Mean in years (SD)	35.8	6.8
**Gender**			
	Male	1	2
	Female	56	98
**Latino**		58	100
**Years Lived in the US**			
	Born in the US	5	9
	More than 10 years	42	75
	10 years or less	9	16
**Educational Attainment**			
	Less than a high school degree	21	39
	High school degree	28	52
	Four-year college degree	4	7
	Graduate degree	1	2
**Household Income**			
	$0–$24,999	46	82
	$25,000–$49,999	9	16
	$50,000–$74,999	0	0
	$75,000+	1	2
**Used SNAP in the Last Year**		19	33
**Number of Children in Household (0–18 years)**			
	1	11	19
	2	31	53
	3	12	21
	4 or more	4	7
**Young Children (0–3 years) in Household**		14	24
**Body Mass Index**			
	Underweight (<18.5)	2	5
	Healthy weight (18.5–24.9)	8	18
	Overweight (25.0–29.9)	12	27
	Obese (30 or above)	22	50
**Language of Survey Administration**			
	Spanish	48	83
	English	6	10
	Both Spanish and English	4	7
**Preferred Language to Speak at Home**			
	Mostly or only English	4	7
	Mostly or only Spanish	40	73
	Equally Spanish and English	11	20
**Ever Seen Toddler Milk in Retail Setting**		53	93
**Number of Times Purchased Toddler Milk**			
	0 times	32	56
	1–2 times	9	16
	3–4 times	2	4
	5–9 times	2	4
	10 or more times	12	21

SNAP: Supplemental Nutrition Assistance Program.

**Table 2 ijerph-18-00528-t002:** Prevalence of themes present in parent interpretations of the “helps support healthy growth” claim on the front of Nido Kinder 1+ toddler milk packaging.

Theme	Description	Exemplary Quote	Prevalence of Theme among Parent Responses *	
			N	%
Growth	Reference to the words grow, growth, or growing or more general references to getting bigger, stronger, or taller	“It helps children to grow.”	21	40
Vitamins, minerals, and nutrients	Reference to the product containing nutrients, vitamins, or minerals	“It helps children to grow healthy and it has many vitamins for children.”	18	35
General development	Reference to general or physical development	“It tells me that it will help my child to have a healthier development.”	7	13
Immunity and illness prevention	Reference to the immune system, immunity, illness, germs, defense, protection, or sickness	“It provides vitamins and nutrients that will prevent your child from having a weak immune system or fragile bones.”	4	8
Bone or muscle	Reference to bone or muscle growth or development	“It helps them to have stronger bones.”	4	8
General health promotion	Reference to the product being healthy, promoting health, or being good for you or for children	“It is healthy for children.”	3	6
Meal or milk substitute	Reference to the product being a solution to picky eating or being used as a substitute for regular milk, breastmilk, or other foods	“Giving extra support that a child might not be getting in their regular meals. Maybe they don’t eat as much fruit or veggies and this might help with that part of their diet.”	3	6
Other ingredients	Reference to other ingredients such as macronutrients (protein, fat, carbohydrates, sugar), probiotics, hormones, additives	“That it contains artificial vitamins to boost children growth, genetically engineered hormones in the milk.”	2	4
Skeptical or misleading	Reference to the claim being misleading or untrue or expression of skepticism about the claim	“The truth is, I don’t think it is healthy to consume it.”	2	4
Brain or cognition	Reference to brain growth or development or cognitive or mental development	“It contributes to children’s bone growth and brain development.”	2	4
Positive perception (not-health related)	Reference to the product quality, generally liking the product or other positive perceptions	“It is good milk for children that are growing.”	1	2

* Data missing from 6 participants. Item wording used in survey: “This product says, ‘Helps support healthy growth.’ What does the phrase “Helps support healthy growth” tell you about the product?”

**Table 3 ijerph-18-00528-t003:** Prevalence of themes present in parent interpretations of “immunity” claim on the front of Nido Kinder 1+ toddler milk packaging.

Theme	Description	Example Quote	Prevalence of Theme among Parent Responses *	
			N	%
Immunity and illness prevention	Reference to the immune system, immunity, illness, germs, defense, protection, or sickness	“Helps them from not getting sick as often or getting stronger in fighting off any type of infection.”	31	67
Vitamins, minerals, and nutrients	Reference to the product containing nutrients, vitamins, or minerals	“It helps boost their immune system by providing key vitamins.”	3	7
Skeptical or misleading	Reference to the claim being misleading or untrue or expression of skepticism about the claim	“I’m not sure that this is true.”	2	4
Growth	Reference to the words grow, growth, or growing or more general references to getting bigger, stronger, or taller	“It helps to prevent sickness. It makes children stronger.”	2	4
General health promotion	Reference to the product being healthy, promoting health, or being good for you or for children	“It strengthens your stomach.”	1	2
General development	Reference to general or physical development	“It can help them to develop more quickly and get sick less I think.”	1	2
Positive perception (not-health related)	Reference to the product quality, generally liking the product or other positive perceptions	“Well I think it is good, I don’t know.”	1	2

* Data missing from 12 participants. Item wording used in survey: “This product says ‘immunity’. What does the word’immunity’ tell you about the product?”

## Data Availability

The datasets generated and/or analyzed during the current study are not publicly available due to IRB restrictions, but are available from the corresponding author on reasonable request.

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
