# Peer review of "Parental Perceptions and Exposure to Advertising of Toddler Milk: A Pilot Study with Latino Parents"

_ijerph, 2021, doi:10.3390/ijerph18020528_

Round 1
Reviewer 1 Report
Title and Abstract
The paper is a well written and interesting brief report that gives some insight into knowledge of toddler milks and their usage in Latino parents. The abstract is a good summary of the reported research. However, the title does not refer to the influence of marketing exposures on parents' perceptions. This is a main focus of the research so should be included in the title if possible e.g. Parental Perceptions and Influence of Advertising on Toddler Milks: A Pilot Study with Latino Parents.
Introduction
The introduction does not make the link to possible adverse health effects associated with high sugar intake i.e. overweight and obesity, dental caries. That point could be made to justify the research. I am concerned that the introduction overly focusses on brands and in particular singles out one brand. This could prejudice the reader and influence objective critique of the research. This could also be seen as an unfair singling out of one product over others. Could this be more general or a reason provided for why the focus is on this one brand?
Methods
These could be clearer. It was not possible to tell exactly how the research had been undertaken and to replicate the procedures from the reporting here. How was this a convenience sample? Were the subjects part of a wider study? If so then it is unclear why participants were reported to be recruited with flyers and by word of mouth. Please clarify if these recruitment methods apply to your sample or the sample of a wider study. If describing the wider study procedures please make this clear and provide the reference where appropriate.
Did the participants in this present study take part in the shopping task? Is it relevant to this study? It seems you are describing the wider study here? Please make this clear. It does not seem relevant to describe activities that were part of the main study and from which findings are not reported.
Please explain why parents needed to consume sugary beverages and alcohol to be included in this study.
Measures
These are not adequately described always. How long did the survey take? Who collected the data? How were the images presented? Why were two brands used here but only one later? Is this because Nido is the only product with added sugar?
How was cognitive testing carried out?
Is reference 23 appropriate/correct?
Analysis
Details are brief. A description of how parents indicated knowledge is needed. Was a scale used as implied in results? Describe the scale and how scored. It would be helpful to state that all data are descriptive only.
Results
Demographics
It would be helpful to know if the sample was representative of the wider Latino population. For example is the prevalence of overweight and obesity similar? Is the high percentage of participants with only a school certificate or lower typical of this population in the US?
Familiarity and Purchase Behaviours
In line with comments in the introduction, it is unclear why this part of the research focusses on a single product. This needs clarification and justification in the introduction. Was information on the range of toddler milks purchased collected? Are Nido and Enfagrow the only products available to this population?
Only 56%(33) participants had bought toddler milks. 51% had bought Nido. does this % refer to the whole sample or to the 56%? It would help to include numbers and percentages when reporting findings of this small sample.
Strengths and limitations are combined. Indicate whether the sample is representative. Table 2 is based on a single product. Therefore findings may not be representative of toddler milks in general.
Perception of claims
Findings are briefly reported and it is necessary to refer to the supplemental table to interpret these.
Conclusions
I believe the conclusion is too strong to make based on this small study. The findings do suggest a lack of knowledge in parents regarding nutritional content of toddler milks and their place (if any) in the young child diet. However, the methods as described, and the analyses which are wholly descriptive are inadequate to base such strong conclusions as policy changes at governmental level. This pilot data is useful to inform a larger study that is more appropriately designed to address the research question.
Author Response
Title and Abstract
The paper is a well written and interesting brief report that gives some insight into knowledge of toddler milks and their usage in Latino parents. The abstract is a good summary of the reported research. However, the title does not refer to the influence of marketing exposures on parents' perceptions. This is a main focus of the research so should be included in the title if possible e.g. Parental Perceptions and Influence of Advertising on Toddler Milks: A Pilot Study with Latino Parents.
Thank you for your feedback. We have updated the title.
Introduction
The introduction does not make the link to possible adverse health effects associated with high sugar intake i.e. overweight and obesity, dental caries. That point could be made to justify the research. I am concerned that the introduction overly focusses on brands and in particular singles out one brand. This could prejudice the reader and influence objective critique of the research. This could also be seen as an unfair singling out of one product over others. Could this be more general or a reason provided for why the focus is on this one brand?
We have added more information about the adverse health effects of added sugar consumption in early childhood (lines 54-58). We chose not to focus exclusively on added sugar intake as not all toddler milk varieties contain added sugar, though most do. The other health concern associated with toddler milk consumption is their potential to displace more nutrient-dense foods (now discussed on lines 58-59) or foods with a variety of flavor and texture in early childhood which are critical for the development of health promoting dietary behaviors (1, 2).
As we now explain in the paper on lines 70-71 and 139-141, our rationale for focusing on Nido is because of their extensive targeted advertising to Latino communities. We have also added information about targeted marketing by other brands to the introduction (lines 72-73).
Methods
These could be clearer. It was not possible to tell exactly how the research had been undertaken and to replicate the procedures from the reporting here. How was this a convenience sample? Were the subjects part of a wider study? If so then it is unclear why participants were reported to be recruited with flyers and by word of mouth. Please clarify if these recruitment methods apply to your sample or the sample of a wider study. If describing the wider study procedures please make this clear and provide the reference where appropriate.
We have made several edits to the paper to clarify the methods (lines 96-109 and lines 114-116). From August to October 2019, we recruited a convenience sample of 61 Latino parents living in North Carolina as part of a pilot study evaluating the impact of sugary drink warnings and taxes on purchases in a naturalistic convenience store laboratory (i.e., the “store pilot study.” This sample answered questions about toddler milks as part of their participation in the store pilot study, as described below. The results of the store pilot study will be reported in a separate publication. As part of the store pilot study, participants attended five weekly study visits in a naturalistic convenience store laboratory, where they completed a shopping task (data to be reported separately) and a self-administered online survey which was programmed using Qualtrics survey software. The toddler milk survey items summarized in this manuscript were included in the self-administered online survey from visit three of the store pilot study.
We describe this as a convenience sample because we used non-probability sampling and enrolled individuals that responded to our in-person recruitment (at bus stops, laundromats, neighborhoods, local non-profits) and that heard about the study from friends and family. We have added more detail to the participants and procedures sections of the methods to clarify our sampling process (lines 96-120).
Did the participants in this present study take part in the shopping task? Is it relevant to this study? It seems you are describing the wider study here? Please make this clear. It does not seem relevant to describe activities that were part of the main study and from which findings are not reported.
We chose to include a short description of the shopping task and incentives in this manuscript in this article so that readers could better understand the main “store pilot study,” especially since that paper has not yet been published. We now explain (lines 114-116) that “The toddler milk survey items summarized in this manuscript were included in the self-administered online survey from visit three of the store pilot study.”
Please explain why parents needed to consume sugary beverages and alcohol to be included in this study.
This was a requirement for the separate “store pilot study” that was evaluating sugary drink warning labels and taxes on purchases in a naturalistic convenience store laboratory. In order to assess reactions to those interventions, it was necessary to have participants that consumed sugary drinks; we now clarify that the eligibility criteria were designed for the purposes of the main “store pilot” study. Participants did not need to purchase alcohol, the needed to have purchased at least one non-alcoholic beverage in the previous week. We have added a sentence to the methods to clarify this further (lines 108-109).
Measures
These are not adequately described always. How long did the survey take? Who collected the data? How were the images presented? Why were two brands used here but only one later? Is this because Nido is the only product with added sugar?
The entire visit three survey took 13 minutes on average. The toddler milk items were one section of that survey. As stated on lines , participants took a self-administered computer survey, programmed using Qualtrics survey software (line 111-116) so the data was collected online using the Qualtrics platform. The images were programmed into the Qualtrics survey and Supplemental table 1 includes which questions displayed images of toddler milk alongside the question text.
Two brands that are commonly advertised on English and Spanish-language TV as well as other outlets were used initially when the definition of toddler milk was presented and with the more general toddler milk items (e.g. familiarity, past purchases, reasons) in order to ensure participants understood that these products are different from infant formula and to understand what products the questions were referring to.
As we now explain in the paper on lines (134-141), we used only an image of Nido in the later items because the items were specific to that product (e.g. what ingredients do you think are in this beverage (results not reported), do you think this beverage has added sugar, what do the claims tell you about this product). Enfagrow Toddler Next Step also contains added sugar, but we selected Nido as opposed to Enfagrow due to their clear emphasis on the Latino market in advertising expenditures.
How was cognitive testing carried out?
We removed mention of the cognitive testing from the manuscript since the toddler milk items were not cognitively tested (this will be reported further in the manuscript with the store pilot results)
Is reference 23 appropriate/correct?
Yes, this is the reference for the paper from which we adapted the survey items about conversations.
Analysis
Details are brief. A description of how parents indicated knowledge is needed. Was a scale used as implied in results? Describe the scale and how scored. It would be helpful to state that all data are descriptive only.
We have included more information on the scales used in the measures section, including the measure of familiarity with toddler milk (line 129). We have also included a sentence that all data are descriptive only due to the small sample size (lines 155-156). The exact item and response option wording can be found in Supplementary Table 1.
Results
Demographics
It would be helpful to know if the sample was representative of the wider Latino population. For example is the prevalence of overweight and obesity similar? Is the high percentage of participants with only a school certificate or lower typical of this population in the US?
We have added a sentence to the limitations describing that due to the use of a small, convenience sample our population is not representative of the US Latino population (line 268). The prevalence of obesity and overweight among Latinos in the US is around 80% compared with 77% in our sample(3). Additionally, the percentage of Latinos in the US with a high school degree or less is about 60% as opposed to 91% in our sample(4).
Familiarity and Purchase Behaviours
In line with comments in the introduction, it is unclear why this part of the research focusses on a single product. This needs clarification and justification in the introduction. Was information on the range of toddler milks purchased collected? Are Nido and Enfagrow the only products available to this population?
Due to limited space in the survey, we presented only two varieties that are commonly marketed on Spanish-language TV(5). The familiarity and purchase behavior questions were asked first generally about toddler milks and then about Nido specifically (Supplemental Table 1). We were interested in Nido specifically given evidence (both peer-reviewed and anecdotal) of Nido’s aggressive targeted marketing to Latino populations. The purchases question asked about toddler milks in general first and then about Nido specifically (“How many times before being in this study have you bought toddler milks for your children?” and “Have you ever purchased this product before being in this study?”). We have included justification as to our focus on these two products in the introduction (lines 70-71) and in the methods section (139-141).
Only 56%(33) participants had bought toddler milks. 51% had bought Nido. does this % refer to the whole sample or to the 56%? It would help to include numbers and percentages when reporting findings of this small sample.
We have added numbers to the percentages for values where the numbers are not presented in another table or figure (lines 173, 182-184, and 213-222)
Strengths and limitations are combined. Indicate whether the sample is representative. Table 2 is based on a single product. Therefore findings may not be representative of toddler milks in general.
We made these changes to our limitations (lines 267-283).
Perception of claims
Findings are briefly reported and it is necessary to refer to the supplemental table to interpret these.
We have added more detail to this section of the results including the text of the question asked of parents related to both claims (lines 189-199).
Conclusions
I believe the conclusion is too strong to make based on this small study. The findings do suggest a lack of knowledge in parents regarding nutritional content of toddler milks and their place (if any) in the young child diet. However, the methods as described, and the analyses which are wholly descriptive are inadequate to base such strong conclusions as policy changes at governmental level. This pilot data is useful to inform a larger study that is more appropriately designed to address the research question.
Thanks for this feedback. We agree and have updated our conclusions accordingly (lines 286-294).
Reviewer 2 Report
Overall, I thought this was a well-written and reasonable paper, albeit with low numbers of participants and untested measures. I would be happy to recommend this for publication subject to minor amendments, detailed below.
Line 31 – The end of the abstract makes a bold call for policy change (which I agree with) but there are limitations to the findings from this research (convenience sample, not geographically diverse, low N, untested measures etc). Could make the call for more research here, or soften the implications drawn from this study.
Line 41 – socioecological determinant such as? Primarily deprivation/inequalities?
Line 46 - What differentiates toddler milk from first formula. The AAP recommendations against toddler milk could also be applied to first formula?
Line 66 – Might need to make clear the categories of claims and that health and nutrition claims are subject to restrictions. Also worth defining explicit and implied claims? Talk about health halo but could be noted that (health and nutrition) claims may be implied- can be effective and circumvent restrictions.
Line 73, space needed before ‘One’.
Line 79 – could delete ‘To address these gaps’. This study addresses one key gap (i.e. does not address identified gap of toddler milk consumption in US).
Line 90 – what was the justification that participants were required to be sugary drink consumer?
Line 133 – I am unsure why high school degree and less have been combined here?
Line 135 – 77% in the table (not 78%)
Line 1142/3 – only 44% of parents had ever purchased toddler milk but 51% had bought Nido?
Fig 1. I’m not sure how balanced the set up to this question was. Were participants given an option to say that they didn’t think parents would want their children to drink toddler milk?
Line 153 and 154 – maybe make clear that participants answered 4 or 5 / 1 or 2, i.e. opposed to an average score of all responses.
Line 188 – suggest being specific (i.e. 44%)
Line 193 stating as opposed to mentioning?
Line 195 – find the health halo term a bit confusing, is it synonymous with implied claims or more about the effect of implied claims?
Line 201 missing ‘the’
Line 206 – could be a little international context, precedent for suggested changes in other countries.
Line 207 – advertisements (missing s)
Line 211 – space before ‘additionally’
Line 213 – double space before ‘In addition’
Line 221 – limitation that participants needed to be computer literate – potentially excluding some low education participants?
Author Response
Reviewer 2:
Overall, I thought this was a well-written and reasonable paper, albeit with low numbers of participants and untested measures. I would be happy to recommend this for publication subject to minor amendments, detailed below.
Thank you for your helpful feedback.
Line 31 – The end of the abstract makes a bold call for policy change (which I agree with) but there are limitations to the findings from this research (convenience sample, not geographically diverse, low N, untested measures etc). Could make the call for more research here, or soften the implications drawn
from this study.
This change has been made (lines 31-33).
Line 41 – socioecological determinant such as? Primarily deprivation/inequalities?
We have added a few examples and citations to this sentence (lines 43-44).
Line 46 - What differentiates toddler milk from first formula. The AAP recommendations against toddler milk could also be applied to first formula?
Toddler milk is intended for children ages 12-36 months; whereas, infant formulas are intended for children from birth to 12 months. There are a variety of differences in both the composition and US regulation of infant formulas and toddler milks, some of which are outlined in this article: Pomeranz JL, Harris JL. Federal Regulation of Infant and Toddler Food and Drink Marketing and Labeling. American journal of law & medicine. 2019;45(1):32-56. The AAP recommendations do not apply to infant formulas. The AAP has a separate set of recommendations around infant feeding in the first year of life which recommend exclusive breastfeeding for the first 4-6 months at which point complementary foods can be introduced alongside continued breastfeeding until 12 months(6).
Line 66 – Might need to make clear the categories of claims and that health and nutrition claims are subject to restrictions. Also worth defining explicit and implied claims? Talk about health halo but could be noted that (health and nutrition) claims may be implied- can be effective and circumvent restrictions.
We have included on lines 77-79 that the predominant category of claims on toddler milks are structure/function claims, and that the claims tested in this study fall within that category (line 137). We have noted that these are not currently required to be approved by FDA prior to use and do not need to be substantiated by scientific evidence.
Line 73, space needed before ‘One’.
This change has been made.
Line 79 – could delete ‘To address these gaps’. This study addresses one key gap (i.e. does not address identified gap of toddler milk consumption in US).
This change has been made.
Line 90 – what was the justification that participants were required to be sugary drink consumer?
This was a requirement for the separate pilot study that was evaluating the impact of sugary drink warning labels and taxes on purchases in a naturalistic convenience store laboratory. In order to assess the impact of those interventions, it was necessary to have participants that consumed some sugary drinks. We used the same sample for this small study as the sugary drink study because our survey questions were asked in the survey after the third shopping task/store visit. We have added more information about this to our methods section on lines 96-109.
Line 133 – I am unsure why high school degree and less have been combined here?
We have changed the way education is presented in the results so the percentage of parents with less than a high school degree and parents with a high school degree are reported separately (lines 161-162).
Line 135 – 77% in the table (not 78%)
This change has been made.
Line 1142/3 – only 44% of parents had ever purchased toddler milk but 51% had bought Nido?
We agree that this is unusual; however, this is what our participants reported. Nido manufactures a variety of products, some of which are fortified milks for older children, so it is possible that parents may have been thinking of other Nido products when responding to this item.
Fig 1. I’m not sure how balanced the set up to this question was. Were participants given an option to say that they didn’t think parents would want their children to drink toddler milk?
This is a good point and an important consideration for future studies on this topic. There was an open-ended option; however, there was not an option stating that parents would not want their children to drink toddler milk. None of the participants’ responses in the open-ended field suggested they thought this would not be something parents would want to provide to their children.
Line 153 and 154 – maybe make clear that participants answered 4 or 5 / 1 or 2, i.e. opposed to an average score of all responses.
This change has been made (lines 184-185).
Line 188 – suggest being specific (i.e. 44%)
This change has been made (line 231).
Line 193 stating as opposed to mentioning?
This change has been made (line 238).
Line 195 – find the health halo term a bit confusing, is it synonymous with implied claims or more about the effect of implied claims?
The health halo effect is a phenomenon whereby shoppers misinterpret a claim about one product attribute as an indication that the entire product is healthy. The claim that causes a health halo effect does not have to be an implied claim but could be a claim categorized and regulated by FDA such as a nutrient content claim like “Good Source of Calcium”, but the claim causes the consumer to assume that because a product contains or lacks (in the case of nutrients like sugar or fat) a vitamin or nutrient that the entire product is healthy. We have added this definition to the discussion in addition to the introduction to minimize confusion (lines 240-241).
Line 201 missing ‘the’
We did not find anywhere to include ‘the’ on the original line 201 (now line 239).
Line 206 – could be a little international context, precedent for suggested changes in other countries.
We are not sure what the reviewer is referring to in this comment.
Line 207 – advertisements (missing s)
This change has been made.
Line 211 – space before ‘additionally’
This change has been made.
Line 213 – double space before ‘In addition’
This change has been made.
Line 221 – limitation that participants needed to be computer literate – potentially excluding some low education participants?
This has been added to our limitations section (lines 276-278).
Reviewer 3 Report
This study, which aims to examine Latino parents' experiences and perceptions relating to purchasing of toddler milk. Given that Latinx children are at a high risk for obesity, this is an interesting study and will significantly contribute to the body of literature. However, there are several comments that should be addressed to improve this manuscript.
1. The introduction section is thorough, although there are times where there was repetition (e.g., "Not recommended, despite these recommendations, etc."). This point felt like it was reiterate in multiple paragraphs. I would suggest the authors review this section to see if there is a way to condense the repetition that occurs with this specific phrase.
2. In the introduction, I would also suggest the authors add in that Latino children have higher rates of childhood obesity that peers of other ethnic groups.
3. This study did not examine parents reasons for purchasing toddler milk; this study asked about parents' perceptions as to why others would purchase toddler milk, which is a key differentiation that needs to be made throughout the paper.
4. Please describe the recruitment methods in more detail. What type of convenience sample was utilized? Are the recruitment methods here the sample for the pilot study from which this study's participants came? More detail is needed to distinguish between recruitment for the pilot study and this study.
5. The insertion point for "Table 1 Supplementary) is off- it seemed like you were stating that Table S1 contained the images, not the survey. Please revise so readers understand the contents of Table S1.
6. Thematic analysis needs a better description of the type of analysis used- inductive coding? Please elaborate here.
7. Table 1 is difficult to read as is. Please different variables (e.g., bolding). Did the survey ask about nationality? I would be interested to see if results varied for those who are immigrant vs. born in the United States. I would suggest stratifying Table 1 by those who did and did not purchase toddler milk, which is a main aim of this study. Without this information, you are not really understanding the sociodemographics of purchasers. This would then need to be added throughout the results and discussion.
8. The title of Figure 1 should be changed- please make it clear that parents were not asked directly why they used toddler milk, but instead why they thought other parents would want their children to drink toddler milk.
9. I am confused about the information presented in Table 2, as it is difficult to determine what the (hg) vs. (i) stands for when there is a percentile associated. I would suggest splitting this into two tables for easier interpretation and clarity.
10. In the discussion, I would mention that a majority of parents had never purchased toddler milk. This is important to take into account when interpreting results.
11. I would discuss why you did not ask reasons why or why not parents purchased toddler milk. Instead, you asked why they thought other parents purchased products. This should be an area of future research since this question was not asked in this study.
12. Instead of saying "the use of some survey items that have not been used in prior studies..." you should state that you used measures that were not valid or reliable, etc. That is a more accurate representation.
Author Response
Reviewer 3:
This study, which aims to examine Latino parents' experiences and perceptions relating to purchasing of toddler milk. Given that Latinx children are at a high risk for obesity, this is an interesting study and will significantly contribute to the body of literature. However, there are several comments that should be addressed to improve this manuscript.
Thank you for your helpful feedback.
- The introduction section is thorough, although there are times where there was repetition (e.g., "Not recommended, despite these recommendations, etc."). This point felt like it was reiterate in multiple paragraphs. I would suggest the authors review this section to see if there is a way to condense the repetition that occurs with this specific phrase.
Thanks for this suggestion. We have revised the introduction to reduce this repetition (lines__-__).
- In the introduction, I would also suggest the authors add in that Latino children have higher rates of childhood obesity that peers of other ethnic groups.
This information has been added (line 76).
- This study did not examine parents reasons for purchasing toddler milk; this study asked about parents' perceptions as to why others would purchase toddler milk, which is a key differentiation that needs to be made throughout the paper.
Thanks for pointing this out. We have revised the language throughout the manuscript to make this clearer (lines 25, 91, 130-132, 178-179, 273-275).
- Please describe the recruitment methods in more detail. What type of convenience sample was utilized? Are the recruitment methods here the sample for the pilot study from which this study's participants came? More detail is needed to distinguish between recruitment for the pilot study and this study.
The samples for the pilot study and this study are identical. For the store pilot study in the naturalistic convenience store laboratory, participants attended five study visits. At each visit, the participants completed an in-person shopping task in the convenience store laboratory and then completed a self-administered online survey programmed into Qualtrics survey software on a computer in the laboratory. The questions about toddler milks were part of the online survey for study visit three. We have added more information to the methods to clarify this (lines 96-120). In the paper, we now explain that we recruited individuals in person (at bus stops, laundromats, neighborhoods, local non-profits) using flyers and by word of mouth until we reached our sample size goal.
- The insertion point for "Table 1 Supplementary) is off- it seemed like you were stating that Table S1 contained the images, not the survey. Please revise so readers understand the contents of Table S1.
This has been moved to lines 138-139.
- Thematic analysis needs a better description of the type of analysis used- inductive coding? Please elaborate here.
We have added more information about our coding process to the analysis subsection of the methods (lines149-153). We utilized an inductive coding process by first reviewing the participants’ responses and then developing codes based on emerging themes.
- Table 1 is difficult to read as is. Please different variables (e.g., bolding). Did the survey ask about nationality? I would be interested to see if results varied for those who are immigrant vs. born in the United States. I would suggest stratifying Table 1 by those who did and did not purchase toddler milk, which is a main aim of this study. Without this information, you are not really understanding the sociodemographics of purchasers. This would then need to be added throughout the results and discussion.
Thanks for this feedback. We have updated Table 1 to improve readability and added bolding of variables. We unfortunately do not have data on nationality, but we do have data on the years our participants have lived in the US. Most of our participants (91%) were not born in the US and a notable proportion (17%) have been in the US for 10 years or less. We added this information to Table 1.
We created a Table 1 with demographics presented for the total sample, purchasers, and non-purchasers (below). We have decided not to include this information in the manuscript as the cell sizes become quite small in some cases and limit our ability to make inferences about characteristics of purchasers and non-purchasers. Moreover, examining predictors of having purchased toddler milk is outside the scope of this pilot study, but we agree it is an important area for future research. We have added this as an area of future research in our discussion (lines 233-235).
Table 1. Participant demographic characteristics overall and among toddler milk purchasers and non-purchasers
|
|
|
|
Overall |
Purchasers(n=25) |
Non-Purchasers (n=32) |
|||
|
|
|
|
N |
% |
N |
% |
N |
% |
|
Age |
|
|
|
|
|
|
|
|
|
|
18-29 years |
|
11 |
19 |
5 |
20 |
6 |
18 |
|
|
30-39 years |
|
30 |
52 |
14 |
56 |
16 |
50 |
|
|
40-49 years |
|
16 |
28 |
6 |
24 |
9 |
28 |
|
|
50+ years |
|
1 |
2 |
0 |
0 |
1 |
3 |
|
|
Mean in years (SD) |
|
35.8 |
6.8 |
34.0 |
7.0 |
36.9 |
6.5 |
|
Gender |
|
|
|
|
|
|
|
|
|
|
Male |
|
1 |
2 |
0 |
0 |
0 |
0 |
|
|
Female |
|
56 |
98 |
25 |
100 |
31 |
100 |
|
Latino |
|
58 |
100 |
25 |
100 |
31 |
100 |
|
|
Years Lived in the US |
|
|
|
|
|
|
|
|
|
|
Born in the US |
|
5 |
9 |
3 |
12 |
2 |
6 |
|
|
More than 10 years |
|
42 |
75 |
16 |
64 |
26 |
84 |
|
|
10 years or less |
|
9 |
16 |
6 |
24 |
3 |
9 |
|
Educational attainment |
|
|
|
|
|
|
|
|
|
|
Less than high school degree |
|
21 |
39 |
8 |
32 |
13 |
45 |
|
|
High school degree |
|
28 |
52 |
15 |
60 |
13 |
45 |
|
|
Four-year college degree |
|
4 |
7 |
2 |
8 |
2 |
7 |
|
|
Graduate degree |
|
1 |
2 |
0 |
0 |
1 |
3 |
|
Household income |
|
|
|
|
|
|
|
|
|
|
$0-$24,999 |
|
46 |
82 |
21 |
88 |
24 |
77 |
|
|
$25,000-$49,999 |
|
9 |
16 |
3 |
12 |
6 |
19 |
|
|
$50,000-$74,999 |
|
0 |
0 |
0 |
0 |
0 |
0 |
|
|
$75,000+ |
|
1 |
2 |
0 |
0 |
1 |
3 |
|
Used SNAP in the last year |
|
19 |
33 |
7 |
28 |
12 |
38 |
|
|
Number of children in household (0-18 yrs) |
|
|
|
|
|
|
|
|
|
|
1 |
|
11 |
19 |
3 |
12 |
7 |
22 |
|
|
2 |
|
31 |
53 |
15 |
60 |
16 |
50 |
|
|
3 |
|
12 |
21 |
5 |
20 |
7 |
22 |
|
|
4 or more |
|
4 |
7 |
2 |
8 |
2 |
6 |
|
Young children (0-3 yrs) in household |
|
14 |
24 |
2 |
6 |
12 |
48 |
|
|
Body Mass Index |
|
|
|
|
|
|
|
|
|
|
Underweight (<18.5) |
|
2 |
5 |
1 |
6 |
1 |
4 |
|
|
Healthy weight (18.5-24.9) |
|
8 |
18 |
5 |
28 |
3 |
12 |
|
|
Overweight (25.0-29.9) |
|
12 |
27 |
3 |
17 |
8 |
32 |
|
|
Obese (30 or above) |
|
22 |
50 |
9 |
50 |
13 |
52 |
|
Language of survey administration |
|
|
|
|
|
|
|
|
|
|
Spanish |
|
48 |
83 |
20 |
80 |
27 |
84 |
|
|
English |
|
6 |
10 |
4 |
16 |
2 |
6 |
|
|
Both Spanish and English |
|
4 |
7 |
1 |
4 |
3 |
9 |
|
Preferred language to speak at home |
|
|
|
|
|
|
|
|
|
|
Mostly or only English |
|
4 |
7 |
2 |
9 |
2 |
6 |
|
|
Mostly or only Spanish |
|
40 |
73 |
15 |
65 |
24 |
77 |
|
|
Equally Spanish and English |
|
11 |
20 |
6 |
26 |
5 |
16 |
|
Ever seen toddler milk in retail setting |
|
53 |
93 |
25 |
100 |
28 |
88 |
|
- The title of Figure 1 should be changed- please make it clear that parents were not asked directly why they used toddler milk, but instead why they thought other parents would want their children to drink toddler milk.
This change has been made.
- I am confused about the information presented in Table 2, as it is difficult to determine what the (hg) vs. (i) stands for when there is a percentile associated. I would suggest splitting this into two tables for easier interpretation and clarity.
We apologize for the confusion. We have created two tables, one for the prevalence of themes in parents’ interpretations of the “Helps Support Healthy Growth” claim and one for the prevalence of themes in parents’ interpretations of the “Immunity” claim. We have also updated the table titles and headings to improve clarity (lines 200-210).
- In the discussion, I would mention that a majority of parents had never purchased toddler milk. This is important to take into account when interpreting results.
We have added this to line 232 in the first paragraph of the discussion.
- I would discuss why you did not ask reasons why or why not parents purchased toddler milk. Instead, you asked why they thought other parents purchased products. This should be an area of future research since this question was not asked in this study.
Thanks for this feedback. We agree with the reviewer that it is more informative to understand parents’ reasons for their own purchases; however, when we developed the items for this study (August 2019), the one article from the US published to date that reports on parent provision of toddler milks had not yet been published (January 2020), so we were unsure if a substantial proportion of parents would report past purchases and be able to provide reasons for their own purchases. We also had limited space in the survey so could not ask for both parents’ own reasons and perceptions of other parents reasons. We instead wanted to understand with this pilot study why parents may be purchasing these products as such little information had been published on the topic at the time. We have added this reasoning to the methods (lines 130-132) and have noted that this is an area that should be explored in further research on lines 273-276.
- Instead of saying "the use of some survey items that have not been used in prior studies..." you should state that you used measures that were not valid or reliable, etc. That is a more accurate representation.
This change has been made (lines 272-273).
References
- Lott M, Callahan E, Welker Duffy E, Story M, Daniels S. Healthy beverage consumption in early childhood: recommendations from key national health and nutrition organizations. Technical scientific report. Durham, NC: Healthy Eating Research, 2019. Available at http://healthyeatingresearch.org.
- Pérez-Escamilla R, Segura-Pérez S, Lott M, on behalf of the RWJF HER Expert Panel on Best Practices for Promoting Healthy Nutrition, Feeding Patterns, and Weight Status for Infants and Toddlers from Birth to 24 Months. Feeding Guidelines for Infants and Young Toddlers: A Responsive Parenting Approach. Durham, NC: Healthy Eating Research, 2017. Available at http://healthyeatingresearch.org.
- Harris JL, Fleming-Milici F, Frazier W, et al. Baby Food F.A.C.T.S: Nutrtition and marketing of baby and toddler food and drinks. Storrs, CT: UConn Rudd Center for Food Policy & Obesity, 2017.
- American Academy of Pediatrics (AAP). Policy statement: breastfeeding and the use of human milk. Pediatrics. 2012;129:e827–e841.
Round 2
Reviewer 3 Report
This revision is much improved from the first draft. Authors have made extensive edits that adequately addressed my comments and the manuscript is now suitable for publication.